| Open Peer Review | Host-Microbial Interactions | Commentary

# Feeling hormonal? Insights into bacterial auxin sensing and its physiological effects

Anne van der Meij[1]

ABSTRACT    Plant–microbe communication involves a rich language of chemical signals. Among these signals are plant hormones such as auxins, which are primarily recognized for their roles in plant development. However, they also function in modulating plant–microbe interactions. Interestingly, many bacteria are capable of producing auxins too. Yet, the mechanisms by which auxins affect bacteria and the regulatory processes controlling their production are largely unknown. Rico-Jiménez and colleagues present new insights into the effects of the auxin indole-3-acetic acid on the physiology of the rhizobacterium *Serratia plymuthica* (M. Rico-Jiménez, Z. Udaondo, T. Krell, and M. A. Matilla, mSystems 9:e00165-24, 2024, https://doi.org/10.1128/msystems.00165-24). Their work provides a deeper mechanistic understanding of bacterial transcriptional responses to plant hormones and the impact on bacterial fitness in the context of the rhizosphere environment.

KEYWORDS    auxin signaling, indole-3-acetic acid, *Serratia plymuthica*, plant–microbe interactions

F eeling stressed is a common experience for humans. But at least, we mostly deal with our own hormones. In contrast, plant-associated microbes may also be exposed to the hormonal cues of their plant hosts. And it is possible that this applies to the rhizobacterium *Serratia plymuthica*, which Rico-Jiménez et al. demonstrate to have substantial physiological responses to the plant hormone auxin (1).

In plants, auxin is important for many processes, including cell division, expansion, and differentiation. But plants do not hold a monopoly on auxin. There are many microbes that metabolize and/or synthesize auxin, thereby interfering with plant developmental processes like altering plant root architecture (2). These microbes greatly affect plant health—for better or for worse (3, 4). As plant–microbe interactions are such a vital determinant for ecological and agricultural outcomes, research has dedicated efforts to identifying microbes that support plant health. *S. plymuthica* is one of those candidates. It is able to synthesize indole-3-acetic acid (IAA), the most common auxin (5, 6). Mutant bacteria that are unable to produce the hormone show an altered transcriptomic profile, indicating that endogenous auxin modulates gene activity (7). Intriguingly, exogenous IAA seems to control *S. plymuthica* antibiotic synthesis by binding the transcription factor AdmX, thereby suppressing the antibiotic synthetic genes (8). These discoveries prompted Rico-Jiménez et al. to further dive into IAA's effects on the bacterium, hoping to better our understanding of the role of IAA in regulatory processes and plant–microbe interactions. It is noteworthy that the concept of auxin affecting bacteria is not new, but reports on the specific pathways through which it exerts its effects on plant-beneficial as well as plant-pathogenic bacteria remain limited (5, 9, 10).

To get started, Rico-Jiménez et al. administered two different concentrations of auxin to *S. plymuthica*, recognizing that effects can vary with concentration. Transcriptional analysis revealed that auxin influenced the expression of genes involved in amino acid

Editor Katrine Whiteson, University of California Irvine, Irvine, California, USA

Ad Hoc Peer Reviewers Barbara N. Kunkel, Washington University in St. Louis, St. Louis, Missouri, USA; Stijn Spaepen, KU Leuven, Leuven, Belgium

Address correspondence to Anne van der Meij, anne.vandermeij@utoronto.ca.

The author declares no conflict of interest.

*The views expressed in this article do not necessarily reflect the views of the journal or ASM.*

See the original article at https://doi.org/10.1128/msystems.00165-24.

transport and metabolism. Consequently, when IAA was present, the rhizobacterium exhibited reduced growth on most amino acids as well as on root exudates, which are rich in amino acids (11). Interestingly, other research by the same group demonstrated that the amino acids tyrosine and phenylalanine induced the gene expression of an enzyme required for IAA production by *S. plymuthica* (7), which could demonstrate a feedback cycle further inhibiting growth. These findings made me wonder whether IAA serves as a signal to slow down bacterial growth, thereby preventing overpopulation. The answer to this question was beyond the scope of the study; however, Rico-Jiménez et al. also show increased motility and inhibited biofilm formation by *S. plymuthica* in the presence of IAA. The absence of biofilm formation suggests that the bacteria do not intend to colonize the IAA source, supporting the broad idea that IAA could have population management functions by driving the bacteria to someplace else.

Besides metabolic and motility changes, Rico-Jiménez et al. show IAA also altered the bacteria's response to multiple threats. For example, IAA-treated cells showed increased antibiotic tolerance, most likely via the induction of various transporter genes. Does auxin induce mild stress, thereby priming the bacterium for other stresses it may encounter in the rhizosphere? The rhizosphere is a competitive place (4), where neighboring microbes secrete all kinds of molecules to give themselves an advantage. Having some protective armor can be quite useful. Yet, nature rarely hands out a silver bullet, because alternatively, Rico-Jiménez et al. demonstrate that phages might profit when *S. plymuthica* is exposed to auxin. Uptake of a previously identified *S. plymuthica* bacteriophage was enhanced on IAA treatment (1), which is likely to result in increased bacterial lysis. Excitingly, this finding may pave the way for new opportunities in combating multidrug-resistant bacteria: phages are promising tools in the fight against disease-causing bacteria, and combined treatments with antibiotics can enhance bacterial killing (12). In this context, the prevalence of auxin-mediated phage killing of pathogens and its potential therapeutic applications are worth further investigation.

Finally, the research culminated with a significant discovery: auxin's ability to bind an additional transcriptional regulator besides the previously identified AdmX regulator, making *S. plymuthica* the first bacterium in which two distinct auxin-binding regulators have been identified. This regulator shares 66% sequence identity with the *Escherichia coli* tryptophan repressor TrpR, which regulates the transcription of genes involved in the tryptophan biosynthetic pathway (13). Notably, TrpR binds to L-tryptophan (L-Trp) with high affinity and to IAA and analogs with low affinity (14), suggesting that the biological relevance of the latter interactions is minimal. However, the *S. plymuthica* variant (TrpR$_{A153}$) binds L-Trp, IAA, and indole-3-pyruvic acid (an intermediate of IAA) with moderate affinity, indicating potential biological relevance. Although the study did not further explore the significance of auxin binding by TrpR$_{A153}$, it does provide a potential molecular explanation linking IAA to transcriptional alterations in amino acid metabolism and the reduced growth phenotype in the presence of amino acids pertinent to the context of the rhizosphere. Moreover, identifying a new transcription factor that binds to IAA is exciting beyond just *S. plymuthica*. The work by Rico-Jiménez et al. could help to identify IAA-responsive transcription factors in other bacteria, including economically relevant pathogens. For example, in the plant pathogen *Pseudomonas syringae* (pv. tomato strain DC3000), IAA alters the expression of over 700 genes (15). Yet, the precise mechanisms by which this occurs remain largely unclear. The work by Rico-Jiménez et al. offers new insights to address this open question.

A substantial body of research has focused on elucidating the microbial inhabitants of plants and their rhizospheres in our quest to understand plant health in relation to their microbiome. However, identifying the microbiome only scratches the surface in terms of understanding its function. Echoing the sentiments of one of my mentors, simply cataloging these microbes is akin to compiling a stamp collection. It is nice to have, but on its own, it is unlikely to give you a deeper understanding of how the postal service works. The crux lies in unraveling the activities of these microbes and their underlying mechanisms, a challenging task because of the complexity of biological systems. The

study by Rico-Jiménez et al. represents a commendable effort into getting insight into the underlying signaling, metabolic, and fitness mechanisms that are at play during plant hormone–microbe interactions. Understanding these interactions is vital for addressing our ecological and agricultural challenges, where the impact of microbes cannot be overstated.

## ACKNOWLEDGMENTS

I would like to thank Kirsten J. Meyer and Eelke D. van den Bos for their valuable input, as well as the reviewers for their insightful feedback.

## AUTHOR AFFILIATION

[1]Department of Biochemistry, Temerty Faculty of Medicine, University of Toronto, Toronto, Canada

## AUTHOR ORCIDs

Anne van der Meij ⓘ http://orcid.org/0000-0003-4174-0112

## ADDITIONAL FILES

The following material is available online.

### Open Peer Review

**PEER REVIEW HISTORY (review-history.pdf).** An accounting of the reviewer comments and feedback.

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
