## [Reviewer comments · mSystems]

Feeling hormonal? Insights into bacterial auxin sensing and its physiological effects

Anne van der Meij

Corresponding Author(s): Anne van der Meij, University of Toronto

Review Timeline:

Submission Date:	June 18, 2024
Editorial Decision:	July 17, 2024
Revision Received:	July 30, 2024
Accepted:	August 13, 2024

Editor: Katrine Whiteson

Reviewer(s): Disclosure of reviewer identity is with reference to reviewer comments included in decision letter(s). The following individuals involved in review of your submission have agreed to reveal their identity: Barbara N. Kunkel (Reviewer #2); Stijn Spaepen (Reviewer #3)

Transaction Report:

DOI: <https://doi.org/10.1128/msystems.00611-24>

Re: mSystems00611-24 (Feeling hormonal? Insights into bacterial auxin sensing and its physiological effects)

Dear Dr. Anne van der Meij:

Thank you for the privilege of reviewing your work. One indication of the quality and excitement surrounding this is that it was very easy to find reviewers. Below you will find my comments, instructions from the mSystems editorial office, and the reviewer comments.

Revision Guidelines

Sincerely,
Katrine Whiteson
Editor
mSystems

Reviewer #1 (Comments for the Author):

This is an interesting, well and pleasantly written commentary on the work by Rico Jiménez and colleagues. As it is right now, however, I am not particularly convinced it provides added value above the original primary research article. I would have liked to read much more about how the original article broadened the current view/scope/vision on the field. I take the TrpR as an example. Why is this so interesting? Apparently binding of auxins by TrpR was demonstrated previously in *E. coli*. What makes

this binding so exciting in this particular case? Another example relates to the phages. What could be the mechanism, the relevance? Are there other examples?

Reviewer #2 (Comments for the Author):

The author does a nice job summarizing the recent study by Rico-Jiménez on the role of auxin in regulating gene expression in the beneficial rhizobacteria *Serratia plymuthica*, and putting the findings into the broader context of what is known about auxin and beneficial plant-microbe interactions. However, the impact of this commentary could be increased by broadening the context a bit more, for example by including mention of recent studies on the roles of auxin in pathogenic plant-microbe interactions. This would be fitting, as in the introductory paragraph a broader reference to plant-associated microbes that impact plant health is made. Further, there seem to be several parallels between the findings of Rico-Jiménez and some recent studies on plant-pathogen interactions (Rico-Jiménez et al, cite some of these in their paper), so readers might find this interesting.

I especially appreciate that the author pointed out several unanswered questions raised by the study, that represent interesting areas for future research. I note one more area of questions that can be added in my specific comments below.

Scientific and editorial comments

- It would be appropriate to reference some more recent reviews on the roles of auxin in plant-microbe interactions. The 2011 Spaepen et al review is excellent but does not cover the more recent advances.
- Line 41: suggested word change: rather than "It made me wonder...." use "These findings made me wonder..."
- Lines 43-45: The statement about increased motility in *S. plymuthica* in the presence of IAA is confusing. My first thought is that increased motility would increase the frequency of interaction, rather than repel it. Some additional explanation would be helpful.
- A point emphasized in the commentary is that the study by Rico-Jiménez showed that IAA binds to the TrpR153 protein, which belongs to a family of transcription factors that are regulated by binding to specific ligand. This is a very exciting discovery, and as the author points out "... provides a potential molecular mechanism strengthening the link between IAA, transcriptional alterations in amino acid metabolism, Noteworthy, the discovery of TrpRA153 probably makes *S. plymuthica* the first bacterium in which two distinct auxin-binding transcriptional regulators have been identified". It is important to note in the commentary that the biological relevance of IAA binding to TrpRA153 was not explored in the study.

Reviewer #3 (Comments for the Author):

The commentary by van der Meij places the recent publication by Rico-Jimenez in a broader perspective to emphasise the importance of the findings by the original research paper. In general several aspects of auxins in microbe-plant interactions are covered and I only have a few (minor) comments that might improve the manuscript.

- 1) Line 8: the term "symbiosis" is used here and for the only time. I would avoid using this term as it require benefits for both partners. As the term is not use later in the perspective, I would keep this more general such as "modulation plant-microbe interactions". in that way, you cover the whole spectrum of this type of interactions.
- 2) Line 17: microbes are constantly exposed to the hormonal cues of their plant hosts. I would be careful with such statement. There is very little evidence that the hormones produced by plants have an influence on (rhizosphere) microbes. First of all, concentrations in plants are very low and no specific exudation has been demonstrated. Effects of hormones on microbes have mainly been demonstrated by external application of the hormone or knock-out out microbial production/biosynthesis as the author describes in the perspective. The main question is whether hormones derived/coming from plants have an effect on microbes; maybe ¹³C-labeling experiments can shed a light on this. In my opinion, it is the same molecules but might be that the "hormonal" influence of plant towards microbes is non-existing. The opposite way (microbe to plant) is well established.
- 3) Lines 46-47: Very general statement and the link to the following sentences is not clear (lines 47 and further). Only from line 50 on it is more clear so I would rephrase lines 46-47.

Reviewer #1 (Comments for the Author):

This is an interesting, well and pleasantly written commentary on the work by Rico Jiménez and colleagues. As it is right now, however, I am not particularly convinced it provides added value above the original primary research article. I would have liked to read much more about how the original article broadened the current view/scope/vision on the field. I take the TrpR as an example. Why is this so interesting? Apparently binding of auxins by TrpR was demonstrated previously in *E. coli*. What makes this binding so exciting in this particular case? Another example relates to the phages. What could be the mechanism, the relevance? Are there other examples?

Thank you for taking the time to review the commentary, your kind words, but mostly the push to broaden the scope of the commentary a bit. I now broadened the scope by including plant-pathogen interactions (line 35, lines 73-78) elaborated more on the significance of the TrpR example (lines 62 – 78), and provided one of the reasons why I think the discovery made around phage susceptibility is interesting (lines 57 – 61). My vision is reflected in the questions / observations that I highlight through the commentary. E.g. lines 43-44, 52-53, 70-73, and the last paragraph. These questions/ views are not represented in the article.

Reviewer #2 (Comments for the Author):

The author does a nice job summarizing the recent study by Rico-Jiménez on the role of auxin in regulating gene expression in the beneficial rhizobacteria *Serratia plymuthica*, and putting the findings into the broader context of what is known about auxin and beneficial plant-microbe interactions. However, the impact of this commentary could be increased by broadening the context a bit more, for example by including mention of recent studies on the roles of auxin in pathogenic plant-microbe interactions. This would be fitting, as in the introductory paragraph a broader reference to plant-associated microbes that impact plant health is made. Further, there seem to be several parallels between the findings of Rico-Jiménez and some recent studies on plant-pathogen interactions (Rico-Jiménez et al, cite some of these in their paper), so readers might find this interesting.

I especially appreciate that the author pointed out several unanswered questions raised by the study, that represent interesting areas for future research. I note one more area of questions that can be added in my specific comments below.

Thank you for taking the time to review my commentary and for providing concrete examples on how to improve it. This helps me to increase the added value of the commentary.

With respect to broadening the context:

I followed up on your suggestion to incorporate pathogenic plant-microbe interactions and the roles of auxin:

- 1) Lines 34/35: It is noteworthy that the concept of auxin affecting bacteria is not new, but reports on the specific pathways through which it exerts its effects in plant-beneficial as well as plant-pathogenic bacteria remain limited
- 2) Lines 73-78: Moreover, identifying a new transcription factor that binds to IAA is exciting beyond just *S. plymuthica*. The work by Jiminez et al. could help to identify IAA-responsive transcription factors in other bacteria, including economically relevant pathogens. For example, in the plant pathogen *Pseudomonas syringae* (pv. tomato strain DC3000), IAA alters the expression of over 700 genes (15). Yet, the precise mechanisms by which this occurs remain largely unclear. The work by Rico-Jiménez et al. offers new insights to address this open question.

I will address the specific scientific and editorial comments below:

Scientific and editorial comments

- It would be appropriate to reference some more recent reviews on the roles of auxin in plant-microbe interactions. The 2011 Spaepen et al review is excellent but does not cover the more recent advances.

Agreed - I added two references:

Kunkel BN, Johnson JMB. 2021. Auxin Plays Multiple Roles during Plant-Pathogen Interactions. *Cold Spring Harb Perspect Biol* 13.

Duca DR, Glick BR. 2020. Indole-3-acetic acid biosynthesis and its regulation in plant-associated bacteria. *Appl Microbiol Biotechnol* 104:8607-8619.

- Line 41: suggested word change: rather than "It made me wonder..." use "These findings made me wonder..."

Agreed – thank you

- Lines 43-45: The statement about increased motility in *S. plymuthica* in the presence of IAA is confusing. My first thought is that increased motility would increase the frequency of interaction, rather than repel it. Some additional explanation would be helpful.

Thank you for pointing this out. I added some additional explanation for my reasoning:

Rico-Jiménez et al. also show increased motility and inhibited biofilm formation of *S. plymuthica* in the presence of IAA. The absence of biofilm formation suggests that the

bacteria do not intend to colonize the IAA source, supporting the broad idea that IAA could have population management functions by driving the bacteria to someplace else.

• A point emphasized in the commentary is that the study by Rico-Jiménez showed that IAA binds to the TrpR153 protein, which belongs to a family of transcription factors that are regulated by binding to specific ligand. This is a very exciting discovery, and as the author points out "... provides a potential molecular mechanism strengthening the link between IAA, transcriptional alterations in amino acid metabolism, Noteworthy, the discovery of TrpRA153 probably makes *S. plymuthica* the first bacterium in which two distinct auxin-binding transcriptional regulators have been identified". It is important to note in the commentary that the biological relevance of IAA binding to TrpRA153 was not explored in the study.

Made clear that the biological relevance was not explored in the study – line 70

Reviewer #3 (Comments for the Author):

The commentary by van der Meij places the recent publication by Rico-Jimenez in a broader perspective to emphasise the importance of the findings by the original research paper. In general several aspects of auxins in microbe-plant interactions are covered and I only have a few (minor) comments that might improve the manuscript.

Thank you for taking the time to review the commentary and pointing out some good points that need adjustment. I'll respond to those below.

1) Line 8: the term "symbiosis" is used here and for the only time. I would avoid using this term as it require benefits for both partners. As the term is not use later in the perspective, I would keep this more general such as "modulation plant-microbe interactions". in that way, you cover the whole spectrum of this type of interactions.

Agreed and adjusted - thank you!

2) Line 17: microbes are constantly exposed to the hormonal cues of their plant hosts. I would be careful with such statement. There is very little evidence that the hormones produced by plants have an influence on (rhizosphere) microbes. First of all, concentrations in plants are very low and no specific exudation has been demonstrated. Effects of hormones on microbes have mainly been demonstrated by external application of the hormone or knock-out out microbial production/biosynthesis as the author describes in the perspective. The main question is whether hormones derived/coming from plants have an effect on microbes; maybe ¹³C-labeling experiments can shed a light on this. In my opinion, it is the same molecules but might be that the "hormonal" influence of plant towards microbes is non-existing. The opposite way (microbe to plant) is well established.

Thank you for pointing this out, because I have to agree that the scientific evidence showing that plant hormones have an effect on plant associated bacteria *in planta* is not overwhelming. My reason for using the statement was to engage the reader in a playful way. I will downplay the statement.

In contrast, plant-associated microbes may also be exposed to the hormonal cues of their plant hosts.

An example that in my opinion allows me to use the downplayed statement to introduce the commentary is work by Djami-Tchatchou et al., (2022) that shows that *P. syringae* genes regulated by IAA *in vitro* were also affected by auxin *in planta*.

When it comes to plant hormone concentrations: in the end we just know very little about their concentration on a micro-scale (which might be relevant when thinking about microbes). If anything, it still adds to your point that the statement was too strong.

3) Lines 46-47: Very general statement and the link to the following sentences is not clear (lines 47 and further). Only from line 50 on it is more clear so I would rephrase lines 46-47.

Removed the statement and added a better introductory sentence for the paragraph (Line 48).

Re: mSystems00611-24R1 (Feeling hormonal? Insights into bacterial auxin sensing and its physiological effects)

Dear Dr. Anne van der Meij:

Your manuscript has been accepted, and I am forwarding it to the ASM production staff for publication. Your paper will first be checked to make sure all elements meet the technical requirements. ASM staff will contact you if anything needs to be revised before copyediting and production can begin. Otherwise, you will be notified when your proofs are ready to be viewed.

Sincerely,
Katrine Whiteson
Editor
mSystems